

# Evaluating the efficacy of curcumin in the management of oral potentially malignant disorders: a systematic review and meta-analysis

Wenjin Shi, Qiuhao Wang, Sixin Jiang, Yuqi Wu, Chunyu Li, Yulang Xie, Qianming Chen and Xiaobo Luo

State Key Laboratory of Oral Diseases, National Clinical Research Center for Oral Diseases, Chinese Academy of Medical Sciences, Research Unit of Oral Carcinogenesis and Management, West China Hospital of Stomatology, Sichuan University, Chengdu, Sichuan, China

## ABSTRACT

**Background:** Oral potentially malignant disorders (OPMDs) not only harbour the risk of malignant transformation but can also affect patients' quality of life owing to severe symptoms. Therefore, there is an urgent need for therapeutic strategies to improve patients' quality of life. The objective of this meta-analysis was to comprehensively assess the efficacy of curcumin in the management of OPMDs.
**Methods:** PubMed, Embase, the Cochrane Library, and Web of Science were searched for clinical trials evaluating the efficacy of curcumin in the treatment of OPMDs from inception until March 2024. RevMan 5.4 software was used to perform statistical and subgroup analyses.
**Results:** Sixteen randomised controlled trials (1,089 patients) were selected. Curcumin exhibited comparable efficacy to conventional controls in alleviating pain ($I^2 = 98\%$, $P = 0.49$) and improving tongue protrusion ($I^2 = 94\%$, $P = 0.51$) in oral submucous fibrosis (OSF). Additionally, topical use of curcumin had an efficacy equivalent to that of conventional therapy in reducing pain ($I^2 = 83\%$, $P = 0.31$) and facilitating clinical remission ($I^2 = 67\%$, $P = 0.38$) of oral lichen planus (OLP).
**Conclusion:** The topical use of curcumin may palliate pain and promote clinical healing in OLP patients. Systemic curcumin can ameliorate the degree of pain and tongue protrusion in OSF. Therefore, our study suggests that curcumin could serve as an alternative treatment for managing OPMDs with lower medical toxicity than steroids, especially when steroids are not suitable. Further studies with larger sample sizes and adequate follow-up periods are required to validate our results.

## INTRODUCTION

Oral potentially malignant disorders (OPMDs) include a spectrum of diseases that involves the oral mucosa presenting an elevated risk of progressing into malignancy, including oral lichen planus (OLP), oral submucous fibrosis (OSF), oral leukoplakia (OLK), proliferative verrucous leukoplakia (*Warnakulasuriya, 2020*). The worldwide

Corresponding authors
Qianming Chen,
qmchen@scu.edu.cn
Xiaobo Luo, luoxbscu@163.com

prevalence of OPMDs is 4.47%, and Asia has the highest incidence of OPMDs compared with that of other geographical locations, which may be associated with risky lifestyle habits in these populations (*Mello et al., 2018*). For instance, the development of OLK and OSF has been consistently related to smoking and areca nut chewing habit, respectively (*Kumari, Debta & Dixit, 2022*).

OPMD patients may experience symptoms such as redness, erosion, ulceration or tingling, and severe painful sensation (*Warnakulasuriya, 2020*). Prolonged oral ulcers or erosion severely impair patients' quality of life owing to the physical and psychological stress caused by the severe pain. This can lead to difficulties in food intake, weight loss, and frequent visits to healthcare providers (*Ryu et al., 2014*; *Liu et al., 2022*; *Wu et al., 2022*). However, improper treatment and management are not only detrimental to patients' quality of life but can also exacerbate the condition and can even be life-threatening.

Despite the emerging diverse treatment strategies in recent years for managing OPMDs, including the systemic or topical application of medicine or non-medical means such as surgery or photodynamic therapy (PDT), varied response rates to different methods for certain OPMDs have been reported, leading to clinical confusion regarding the most appropriate treatment approach for OPMDs (*Dionne et al., 2015*; *Binnal et al., 2022*). Among these medical approaches, topical corticosteroids have shown considerable advantages and are the first line of treatment for OLP (*Lodi et al., 2020*). Steroids are routinely used to manage OSF (*Warnakulasuriya & Kerr, 2016*). However, long-term steroid application may induce severe oral adverse effects such as mucosal redness, atrophy, and secondary candidiasis, which restrict their usage (*Meena et al., 2017*; *Lodi et al., 2020*; *Khosrojerdi et al., 2023*). PDT has reportedly demonstrated significant therapeutic potential for OLK; however, patients may be reluctant to undergo PDT because of its invasiveness. A systematic review indicated that PDT may cause side effects, including burning sensation, itching, and tingling (*Li et al., 2019*). Thus, there is an urgent need to identify an ideal substitute for managing OPMDs, particularly those with fewer adverse effects.

Curcumin is a yellow substance of the ginger family extracted from *Curcuma longa*. In addition to curcumin, there are other components in turmeric, such as demethoxycurcumin and bisdemethoxycurcumin, which account for approximately 5% of the total components of *Curcuma longa* and are the main metabolic forms of curcumin in the body (*Kocaadam & Şanlier, 2017*). Curcumin exerts anti-inflammatory effects by lowering circulating concentrations of pro-inflammatory biomarkers (C-reactive protein, interleukins-6, interleukins-8, tumour necrosis factor-$\alpha$, and monocyte chemoattractant protein-1 concentrations) and increasing levels of anti-inflammatory mediators (interleukin-10) (*Sahebkar et al., 2016*; *Derosa et al., 2016*; *Ferguson, Abbott & Garg, 2021*). Additionally, curcumin exhibits antioxidant, antifibrinolytic, analgesic, and anticancer effects (*Smith et al., 2010*; *Sahebkar & Henrotin, 2016*; *Giordano & Tommonaro, 2019*), enabling its wide application with systemic means in the therapy against tumours, inflammatory diseases, and various systemic diseases such as autoimmune disorders. Notably, curcumin has shown efficacy in the treatment of oral diseases *via* both systemic and topical means, such as controlling signs and symptoms of oral mucositis and

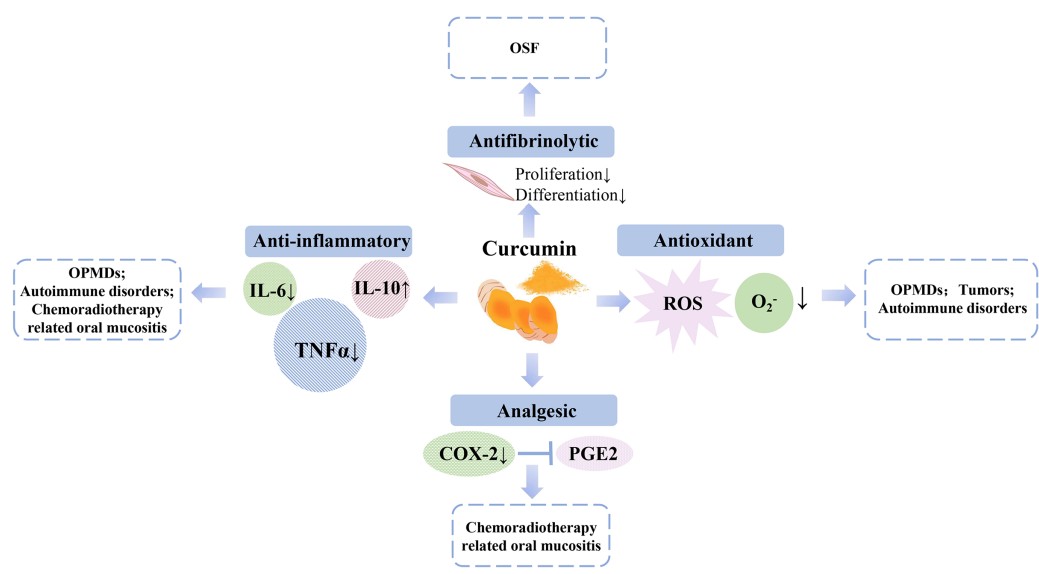

**Figure 1 The potentially diverse mechanisms of curcumin for managing diseases of oral cavity.**

improving OSF and OLP (*White, Chamberlin & Eisenberg, 2019*; *Al-Maweri, 2019*; *Normando et al., 2019*) (Fig. 1). Moreover, several studies have indicated that both systemic and topical applications of curcumin rarely exhibit apparent adverse effects (*Nosratzehi et al., 2018*; *Piyush et al., 2019*), and its side effects are usually dose-dependent and mild (*Khosrojerdi et al., 2023*).

To date, several randomised controlled trials (RCTs) have indicated that curcumin plays a crucial role in improving the symptoms of OPMDs, the mechanisms of which are mainly attributed to immune regulation and inflammation control. Based on the original studies, several systematic reviews and meta-analyses have been conducted, and varying results have been reported (*Ara et al., 2016*; *Al-Maweri, 2019*; *White, Chamberlin & Eisenberg, 2019*; *Rai et al., 2021*, *2023*; *Moayeri et al., 2024*; *Shao, Miao & Wang, 2024*). Therefore, it is necessary to summarise the effects of curcumin in the management of OPMDs. The objective of this systematic review and meta-analysis was to obtain more comprehensive and reliable results. The results of our study could potentially provide better evidence for the clinical selection of curcumin and guide the the design of future clinical trials.

# MATERIALS AND METHODS

## Protocol and registration

The present systematic review and meta-analysis was conducted in accordance with the PRISMA 2020 guidelines (*Page et al., 2021*). The protocol was submitted to the International Prospective Register of Systematic Reviews (PROSPERO) register (CRD42022378624) to ensure low bias, high accuracy, completeness, and transparency.

## Search strategy

Four databases—PubMed, Embase, Cochrane Library, and Web of Science—were searched to obtain all relevant trials related to the management of OPMDs by curcumin. All clinical

trials published before 21 March 2024, with no time period limitations, were screened. To identify additional articles, we manually searched the reference lists of all the relevant meta-analyses. The following key words were used: 'curcumin' or 'turmeric' or 'curcuma longa' or 'diferuloylmethane' and 'oral potentially malignant disorders' or 'oral precancer' or 'oral premalignant lesions' or 'oral potentially malignant lesions' or 'oral submucous fibrosis' or 'oral leukoplakia' or 'proliferative verrucous leukoplakia' or 'oral erythroplakia' or 'oral lichen planus' or 'discoid lupus erythematosus' or 'actinic cheilitis' (Table S1).

## Inclusion criteria

Population (P): human participants diagnosed clinically and histologically with OLP (to differentiate between OLP and oral lichenoid lesions) and clinically and/or histologically with the other OPMDs mentioned above.

Intervention (I): systemic or topical curcumin or curcuma preparation.

Comparison (C): conventional therapy (curcumin-free) or placebo.

Outcome (O): signs and symptoms of OPMDs (pain, mouth opening, tongue protrusion, lesion size, and histological changes); secondary outcome: adverse effects.

Study design (S): randomized controlled trials (RCT), cohort and case-control studies.

## Exclusion criteria

(1) Uncontrolled trials or those combining curcumin with other medication to treat OPMDs.
(2) Case reports, reviews, meta-analyses, letters to the editor, *in vitro* studies, and animal studies.
(3) Participants having other systemic diseases or drug-induction.
(4) Studies with insufficient baseline or outcome indicators for calculation.
(5) Unavailable full text or non-English language articles.

## Screening process

All relevant studies were exported to the Zotero reference management software, and duplicate studies were eliminated. Irrelevant studies were excluded after reading their titles and abstracts. Two reviewers (W.J.S. and Q.H.W.) thoroughly assessed the full texts of the remaining studies to screen relevant literature, and disagreements were resolved by a third reviewer (S.X.J.).

## Data extraction

Date extraction was independently performed by two reviewers (W.J.S. and Q.H.W.), and the following data were extracted: (1) country and authors of the study; (2) type of OPMDs; (3) study design; (4) number, sex ratio and mean age of participants included in the study; (5) diagnostic methods; (6) interventions in treatment and control groups; (7) evaluation indicators; (8) duration and follow-up; (9) adverse effects; and (10) main results.

### Dealing with missing data

Owing to missing reports of standard deviations in some of the included studies, we attempted to contact these trial authors for clarification and to provide data for unreported outcomes. Correlation coefficients were utilised to calculate and impute the change in standard deviation from baseline according to the Cochrane Handbook for Systematic Reviews of Interventions (*Higgins & Green, 2008*).

### Risk of bias assessment

The risk of bias was assessed independently by two reviewers (W.J.S. and Q.H.W.) adopting the risk-of-bias tool for RCT recommended by the Cochrane Handbook (RoB2; Cochrane Collaboration, London, UK), including the domains of bias, signalling questions, and risk of bias.

### Data synthesis

The meta-analysis was performed using Review Manager (version 5.4, Cochrane Collaboration, Oxford, UK). The standardised mean difference (SMD) and 95% confidence interval (CI) were computed, considering that all outcomes were continuous variables. Heterogeneity was analysed using $I^2$. When significant heterogeneity was present among the included trials ($I^2 > 50\%$, $P \leq 0.1$), a random-effect model was applied, and the source of heterogeneity was further investigated; otherwise, a fixed-effect model was adopted.

### Analysis of subgroups or subsets

Subgroup analyses were performed on the basis of drug administration (topical or systemic). The $I^2$ statistic was used to measure the heterogeneity among subgroups in each analysis. Subgroup analyses were employed to clarify heterogeneity when substantial heterogeneity (>50%) was observed.

## RESULTS

### Study selection

Figure 2 shows the process and results of the study selection. A total of 557 articles were selected through database and citation searches, of which 359 were removed because of duplication. Among the remaining 198 studies, 142 were excluded owing to irrelevant titles and abstracts. Of the remaining 56 studies, five were excluded because of the inability to obtain the full text. After full-text screening of the 51 remaining studies, 35 were excluded for various reasons (Table S2). Ultimately, 16 studies were included in the final analysis.

### Description of the included studies

Only 16 RCT involving three types of OPMDs (OSF, OLP, and OLK) from 2014 to 2023 were included (Table 1), whereas numerous trials evaluating the combined efficacy of curcumin with other medications for OPMDs were excluded (Table S2). No studies on proliferative verrucous leukoplakia, oral erythroplakia, actinic cheilitis, or discoid lupus

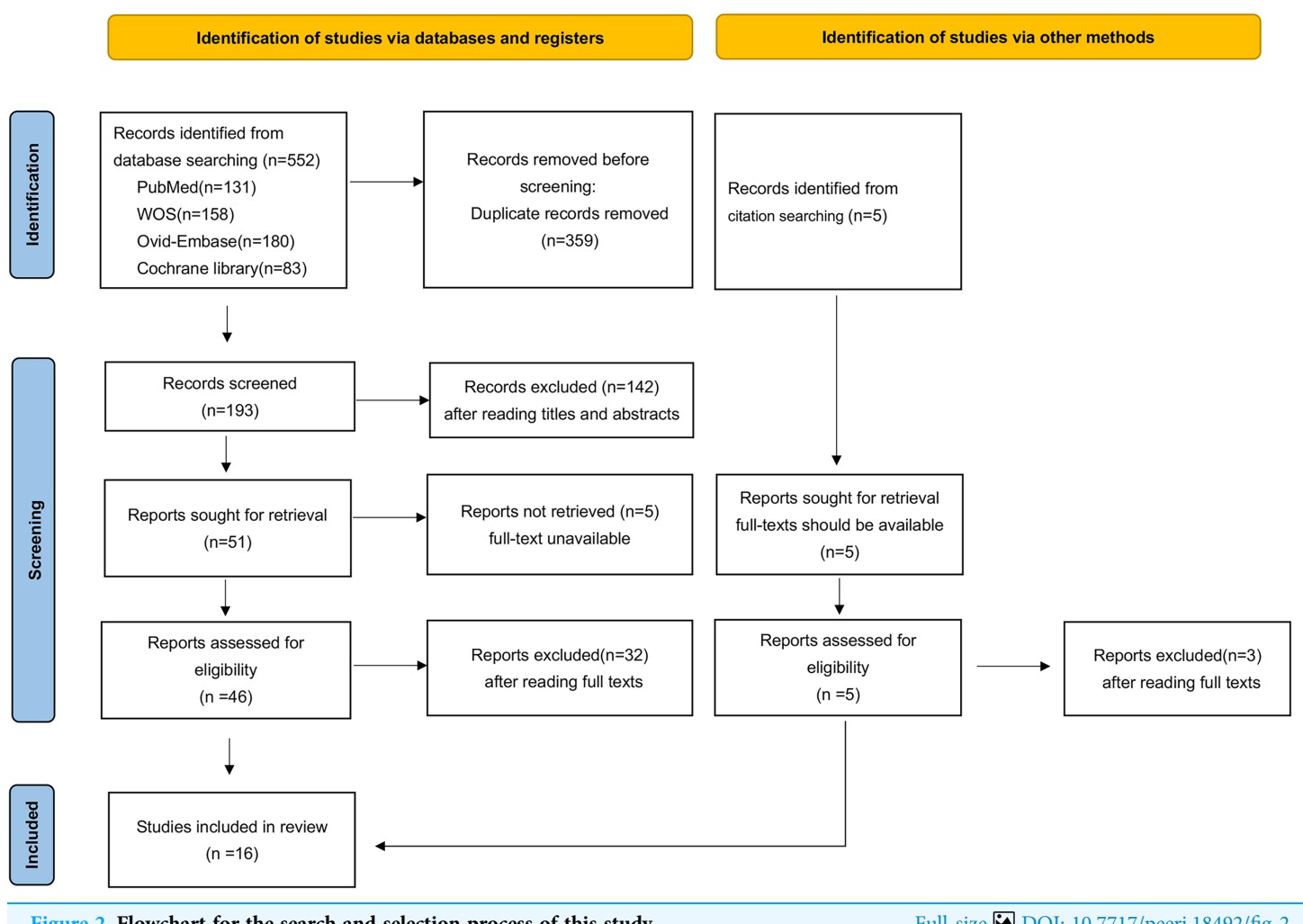

**Figure 2 Flowchart for the search and selection process of this study.**

erythematosus were found, and cohort and case-control studies were excluded because they were uncontrolled trials (*Chainani-Wu, Collins & Silverman, 2012*; *Naik et al., 2019*). A total of 1,089 patients from India, Iran, Nepal, and the United States were enrolled in the study. Further details based on each individual OPMD are provided below.

## OSF

Eleven studies involving 675 participants reported the therapeutic effects of curcumin on OSF (*Yadav et al., 2014*; *Hazarey, Sakrikar & Ganvir, 2015*; *Saran et al., 2018*; *Ara et al., 2018*; *Piyush et al., 2019*; *Rai et al., 2019*; *Srivastava et al., 2021*; *Nerkar Rajbhoj et al., 2021*; *Adhikari et al., 2022*; *Namratha et al., 2023*; *Upadhyay et al., 2023*). OSF was diagnosed both clinically and histologically in five studies (*Yadav et al., 2014*; *Ara et al., 2018*; *Piyush et al., 2019*; *Nerkar Rajbhoj et al., 2021*; *Namratha et al., 2023*) and only clinically in the remaining six studies (*Hazarey, Sakrikar & Ganvir, 2015*; *Saran et al., 2018*; *Rai et al., 2019*; *Srivastava et al., 2021*; *Adhikari et al., 2022*; *Upadhyay et al., 2023*).

Curcumin was administered systemically (as capsules or tablets) in 10 included studies and topically (as a gel) in the remaining one study. The drugs in the control group included

**Table 1 General characteristics of the included literatures in this systematic review and meta-analysis.**

| Study | Type of OPMDs and diagnostic approach | Study design | Experimental group | | | Control group | | | Outcome accessed | Follow-up | Adverse effects | Main results |
|---|---|---|---|---|---|---|---|---|---|---|---|---|
| | | | Sample size | M/F; Age mean/range | Formulation | Sample size | M/F; Age mean/range | Formulation | | | | |
| *Namratha et al. (2023)* India | OSF, Diagnosis clinically and histologically | RCT | 14 | NI | Curcumin: 400 mg twice daily for 3 months | 14 | NI | Placebo: twice-daily doses of placebo capsules | Mouth Opening (MO), Numerical rating scale (NRS) | 3M | NI | When employed as a combination therapy in the initial management of people with OSMF, curcumin can help patients with their clinical symptoms. |
| *Upadhyay et al. (2023)* India | OSF, Diagnosis clinically | RCT | 30 | Age range: 18–50 | Curcumin lozenges: 2 g daily | 30 | Age range: 18–50 | Tenovate™ (topical clobetasol propionate): three times daily | MO, Burning Sensation (visual analog scale, VAS) | 6W (Recalls were done every week for up to 6 weeks) | No side effects | There was a significant improvement in mouth opening with clobetasol group and nonsignificant results were obtained with curcumin group. |
| *Adhikari et al. (2022)* Nepal | OSF, Diagnosis clinically | RCT | 17 | M/F: 13/4 Mean age: 36.06 | Curcumin: 2 gm daily in four divided dosage for 6 weeks | 17 | M/F: 11/6 Mean age: 34.53 | Placebo: 2 gm daily in four divided dosage for 6 weeks | VAS, MO, Tongue Protrusion (TP), Cheek Flexibility (CF) | 3M (Patients were evaluated at baseline and, 6th, 8th, 12th week) | No reported side effects | Curcumin in combination with baseline treatment of intralesional dexamethasone is efficacious in the treatment of OSF. |
| *Nerkar Rajbhoj et al. (2021)* India | OSF, Diagnosis clinically and histologically | RCT | 30 | M/F: 28/2 Mean age: 29.4 | Curcumin gel: apply 5 mg of gel at respective site for 3–4 times a day | 30 | M/F: 27/3 Age mean: 31.2 | Aloe Vera gel: same procedure | Burning Sensation Score (VAS); MO | 1M (Patients were evaluated every 2 weeks for 6 weeks) | No side effects | Curcumin gel and Aloe Vera gel are effective in improving OSMF symptoms, but Aloe Vera gel is more efficacious in burning sensation improvement without any side effects. |
| *Srivastava et al. (2021)* India | OSF, Diagnosis clinically | RCT | 40 | M/F: 71/9 Mean age: 33.5 Age range: 31–40 | Curcumin lozenges: three times daily for 3 months. | 40 | M/F: 71/9 Mean age: 33.5 Age range: 31–40; | Intralesional infiltration: 2 mL dexamethasone twice a week for 3 months | VAS, MO, TP | 3M (Patients were evaluated every month) | None of the participants reported with any side effects | There was significant clinical improvement in mouth opening and subjective symptoms, like burning sensation/pain associated with the lesion and tongue protrusion in the experiment group as compared to control group. |

(Continued)

| Study | Type of OPMDs and diagnostic approach | Study design | Experimental group Sample size | Experimental group M/F; Age mean/range | Experimental group Formulation | Control group Sample size | Control group M/F; Age mean/range | Control group Formulation | Outcome accessed | Follow-up | Adverse effects | Main results |
|---|---|---|---|---|---|---|---|---|---|---|---|---|
| Piyush et al. (2019) India | OSF, Diagnosis clinically and histologically | RCT | 30 | M/F: 70/20 Mean age: 32 Age range: 17–60 | Curcumin tablet: 300 mg twice daily for 6 months | C1: 30; C2:30 | M/F: 70/20 Mean age: 32 Age range: 17–60 | C1:Lycopene capsules: 8 mg twice daily for 6 months; C2: Placebo capsules once daily for 6 months | VAS, MO, TP, CF | 9M (Patients were evaluated every month) | There was no report of side effects | Curcumin and lycopene are equally effective in the management of both subjective and objective symptoms of OSF. |
| Rai et al. (2019) India | OSF, Diagnosis clinically | RCT | 40 | M/F: 6/1 Mean age: 33.43 | Turmix tablet: 3 times per day for 12 weeks. | 39 | M/F: 6/1 Mean age: 33.43 | Antioxidant table: twice daily for 12 weeks | VAS, MO, TP | 3M (Patients were evaluated every 2 weeks) | Facial flushing and erythema of the palms; discomfort, nausea and stomach upset | Curcumin has a very promising role in the management of OSMF and could emerge as an alternative to antioxidants. |
| Ara et al. (2018) India | OSF, Diagnosis clinically and histologically | RCT | 50 | M/F: 45/5 Mean age: 25.44 | Curcumin capsules: 500 mg per capsule, take two capsules per day | 50 | M/F: 48/2 Mean age: 25.42 | Placebo capsules: two capsules per day | VAS, MO, TP, CF | 6M (Patients were evaluated at baseline and 6th month) | No reported side effects | Patients in experiment group showed statistically significant improvement in all the subjective signs and symptoms and histopathological changes. |
| Saran et al. (2018) India | OSF, Diagnosis clinically | RCT | 30 | M/F: 28/2 Mean age: 27.90 | Curcumin: 300 mg daily for 3 months | 30 | M/F: 28/2 Mean age: 26.00 | Lycopene: orally given 4 mg in two divided doses per day for a period of 3 months | VAS, MO | 3M (Patients were evaluated every 15 days) | No reported side effects | The present study revealed that lycopene is better than curcumin in improving mouth opening and both the medication showed a beneficial effect on reducing the symptoms of OSF. |
| Hazarey, Sakrikar & Ganvir (2015) India | OSF, Diagnosis clinically | RCT | 15 | Age range: 18–50 | Longvida lozenges (400 mg lozenges): the total daily dose decided was 2 g | 15 | Age range: 18–50 | Topical clobetasol propionate–enovate: three times daily | VAS, MO | 9M (Patients were evaluated at baseline and 3rd, 6th, 9th month) | No reported side effects | It is evident from the study that curcumin holds good promise in the treatment of OSF in future. |
| Yadav et al. (2014) India | OSF, Diagnosis clinically and histologically | RCT | 20 | M/F: 15/5 Mean age: 37.9 | Curcumin tablets: two tablets (Turmix 300 mg) per day for 3 months | 20 | M/F: 16/4 Mean age: 40.8 | Dexamethasone & 1,500 I.U Hyaluronidase: weekly intralesional injection of 4 mg | VAS, MO, TP | 3M (Patients were evaluated every month) | No side effects were reported by any patient. | Turmix is beneficial and effective in reducing burning sensation in early OSF patients. |

| Study | Type of OPMDs and diagnostic approach | Study design | Experimental group Sample size | M/F; Age mean/range | Formulation | Control group Sample size | M/F; Age mean/range | Formulation | Outcome accessed | Follow-up | Adverse effects | Main results |
|---|---|---|---|---|---|---|---|---|---|---|---|---|
| Kia et al. (2020) Iran | OLP, Diagnosis clinically and histologically | RCT | 29 | M/F: 4/25 Mean age: 51.86 | Nano-Curcumin: 80 mg after breakfast | 28 | M/F: 5/23 Mean age: 53.67 | Prednisolone: 10 mg, after breakfast | VAS, Thongprasom scale, Lesion size | 1M (Patients were evaluated at baseline and 1st, 2nd, 4th week) | No reported side effects | The results have shown that oral Curcumin can be used as an alternative therapy for OLP in patients with the contraindicated Corticosteroids. |
| Nosratzehi et al. (2018) Iran | OLP, Diagnosis clinically and histologically | RCT | 20 | M/F: 9/11 Age range: 28–60 | Curcumin pates: three times a day after meals | 20 | M/F: 5/15 Mean age: 38.5 Age range: 28–60 | Betamethasone local steroid lotion (0.1%): three times daily and nystatin suspension | VAS and Lesion size | 3M (Patients were evaluated at 1st, 2nd, 4th, 8th, 12th week) | No side effects were observed | Curcumin was effective in the treatment of oral lichen planus lesions and resulted in decreases in lesion sizes, pain and burning sensation severities and changes in classification of the lesions without any complications. |
| Thomas et al. (2017) India | OLP, Diagnosis clinically and histologically | RCT | 19 | M/F: 19/56 Age range: 20–70 | Curcumin oral gel: thrice daily | 25 | M/F: 19/56 Mean age: 20–70 | Triamcinolone acetonide (0.1%): thrice daily, dose was tapered accordingly | NRS, MOMI | 3M (Patients were evaluated every 2 weeks) | Curcumin poses negligible adverse effects | Curcumin oral gel can bring about clinical improvements in OLP patients, it can be used as a maintenance drug after the patient is treated with an initial course of corticosteroids. |
| Kia et al. (2015) Iran | OLP, Diagnosis clinically and histologically | RCT | 25 | M/F: 10/15 Mean age: 49.24 | Curcumin paste (5%): three times a day for 4 weeks | 25 | M/F: 4/21 Mean age: 52.08 | Triamcinolone paste (0.1%): three times a day for 4 weeks | VAS, Appearance score (Thongprasom criteria) | 1M (Patients were evaluated every 2 weeks) | Insignificant side effects. | Application of topical curcumin can be suggested for treatment of OLP because of its desirable antiinflammatory effects and insignificant side effects. |
| Kuriakose et al. (2016) India | OLK, Diagnosis clinically and histologically | RCT | 111 | M/F: 79/32 Mean age: 54 Age range: 40–74 | Curcumin capsules: 600 mg in a twice-daily regimen consumed after food (3.6 g/day) for 6 months | 112 | M/F: 82/30 Mean age: 55 Age range: 26–74 | Placebo capsules: identical in physical characteristics and dispensed similar to the curcumin | Clinical response, histologic response, combined clinical and histologic response | 12M (Patients were evaluated at baseline and 6th, 12th month) | Anaemia, skin/ subcutaneous tissue disorders, and hypertension | Combined clinical and histologic response assessment indicated significantly better response with curcumin |

**Notes:**
Abbreviations: RCT, randomized control trial; F/M, Female/Male; NI, non-informed; M, month(s); W, week(s); OSF, oral submucous fibrosis; OLP, oral lichen planus; OLK, oral leukoplakia.
Pain: VAS (patients ranked the severity of pain on a 10-cm horizontal line graded from 0 to 10; 0 indicated no pain and 10 indicated the most severe pain) and NRS (asking the patients to assign a numerical score representing the intensity of their burning sensation on the scale from 0 to 10, with 0 being no burning sensation and 10 being worst imaginable burning sensation) are both range from 0–10 and thus combined in the results section.
Mouth opening: The distance between the maxillary and mandibular central incisor tooth was measured by vernier calipers in millimeter.
Tongue protrusion: The distance between the tip of the tongue at maximum extension and the maxillary mesial incisors-tongue contact point was measured in millimeter.
Lesion size and histologic changes: Lesion size was calculated by multiplying the longest vertical diameter by the largest horizontal diameter; Histologic complete response, complete reversal of dysplasia/hyperplasia to normal epithelium; histologic partial response, regression of the degree of dysplasia; histologic stable disease, no change in the degree of dysplasia; and histologic progressive disease, any increase in severity grade.

placebo capsules (*Ara et al., 2018*; *Piyush et al., 2019*; *Adhikari et al., 2022*; *Namratha et al., 2023*), intralesional injection of corticosteroids (*Yadav et al., 2014*; *Srivastava et al., 2021*), and other active treatments such as lycopene or antioxidant agents (*Hazarey, Sakrikar & Ganvir, 2015*; *Saran et al., 2018*; *Rai et al., 2019*; *Nerkar Rajbhoj et al., 2021*; *Upadhyay et al., 2023*). The follow-up duration ranged from 6 weeks (*Nerkar Rajbhoj et al., 2021*; *Upadhyay et al., 2023*) to 9 months (*Hazarey, Sakrikar & Ganvir, 2015*; *Piyush et al., 2019*).

Regarding the outcome measures reported in the included studies, all studies assessed mouth opening. Ten included studies applied the visual analogue scale (VAS) scoring system, and one applied a numerical rating scale (NRS) (*Upadhyay et al., 2023*) to assess burning sensation and pain. Six studies evaluated changes in tongue protrusion ability (*Yadav et al., 2014*; *Ara et al., 2018*; *Piyush et al., 2019*; *Rai et al., 2019*; *Srivastava et al., 2021*; *Adhikari et al., 2022*), and cheek flexibility was assessed in three studies (*Ara et al., 2018*; *Piyush et al., 2019*; *Adhikari et al., 2022*).

### OLP

Four studies (*Kia et al., 2015*, *2020*; *Thomas et al., 2017*; *Nosratzehi et al., 2018*) reported the effect of curcumin on 201 OLP patients aged 38.5–70 years. Either gel or paste of curcumin was topically applied three times a day in three studies (*Kia et al., 2015*; *Thomas et al., 2017*; *Nosratzehi et al., 2018*), and daily usage of an 80 mg curcumin capsule was reported in one study (*Kia et al., 2020*). Heterogeneity was observed in the administration of medication within the control sets. The comparative drugs used were 0.1% triamcinolone paste in two studies (*Kia et al., 2015*; *Thomas et al., 2017*) and betamethasone local steroid lotion in one study (*Nosratzehi et al., 2018*), both of which were applied three times daily to the lesion. Prednisolone (10 mg) was administered daily during the whole treatment period in one study (*Kia et al., 2020*). The follow-up period of the included studies ranged from 1 to 3 months.

Burning sensation/pain was the most frequently measured outcomes; three studies assessed the outcomes using VAS (*Kia et al., 2015*, *2020*; *Nosratzehi et al., 2018*) and one was based on the NRS (*Thomas et al., 2017*). The Thongprasom criteria were utilised in two studies to assess the degree of improvement in clinical signs and symptoms (*Kia et al., 2015*, *2020*). In addition to the aforementioned studies, one study assessed lesion size (*Nosratzehi et al., 2018*) and one further evaluated signs (erythema and ulceration) using the modified oral mucositis index (*Thomas et al., 2017*).

### OLK

One study involving 223 participants assessed the efficacy of curcumin in OLK (*Kuriakose et al., 2016*). The patients were clinically and histologically diagnosed with OLK. Curcumin (900 mg) capsules were used twice daily, whereas the placebo capsules were divided into two doses per day in the control group. After 6 months of treatment, clinical and histopathological evaluations were performed.

### Risk of bias of the included studies

Among the 16 RCTs, the bias was high in two studies (*Nosratzehi et al., 2018*; *Kia et al., 2020*), low in six studies (*Chainani-Wu et al., 2012*; *Nosratzehi et al., 2018*;

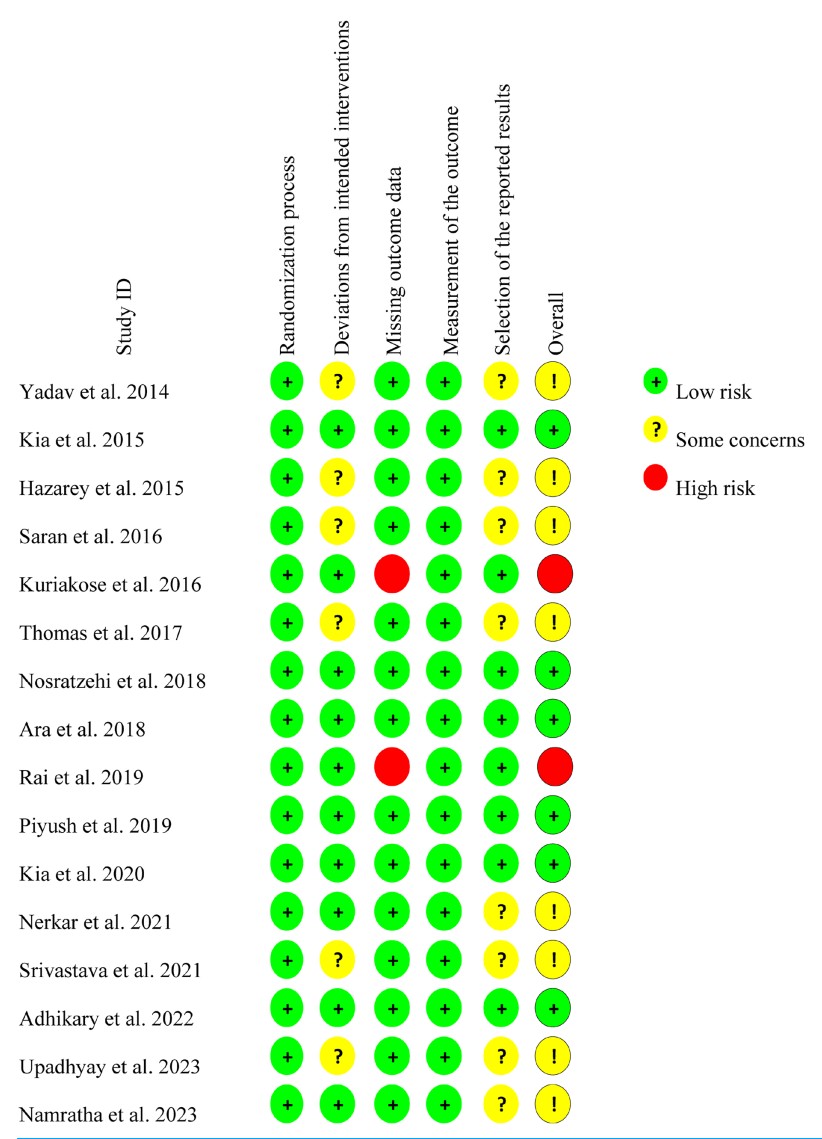

**Figure 3 Critical appraisal about the study quality of the included randomized controlled trials** (*Yadav et al., 2014*; *Kia et al., 2015*; *Hazarey, Sakrikar & Ganvir, 2015*; *Saran et al., 2018*; *Kuriakose et al., 2016*; *Thomas et al., 2017*; *Nosratzehi et al., 2018*; *Ara et al., 2018*; *Rai et al., 2019*; *Piyush et al., 2019*; *Kia et al., 2020*; *Nerkar Rajbhoj et al., 2021*; *Srivastava et al., 2021*; *Adhikari et al., 2022*; *Upadhyay et al., 2023*; *Namratha et al., 2023*).

*Piyush et al., 2019*; *Kia et al., 2020*; *Adhikari et al., 2022*), and of 'some concerns' in eight studies (*Yadav et al., 2014*; *Hazarey, Sakrikar & Ganvir, 2015*; *Thomas et al., 2017*; *Saran et al., 2018*; *Srivastava et al., 2021*; *Nerkar Rajbhoj et al., 2021*; *Namratha et al., 2023*; *Upadhyay et al., 2023*). Additionally, the "Selection of the reported results" domain was the main source of bias (Fig. 3), which may have been caused by the lack of a prespecified analysis plan (PROSPERO) mentioned in texts. Moreover, the 'Deviations from Intended Interventions' domain also introduced bias owing to differences in drug formulations between control and experiment groups, which undermined the effectiveness of the

double-blind. The 'High risk' designation was owing to the 'Selection of the Reported Result', which stemmed from the absence of follow-up data.

## Qualitative findings

### OSF

Four studies reported that curcumin was significantly more robust than the placebo in improving all or most of the assessed outcomes (*Ara et al., 2018*; *Piyush et al., 2019*; *Adhikari et al., 2022*; *Namratha et al., 2023*). In the remaining seven studies, curcumin was comparable to control groups in some of the assessed outcomes (*Yadav et al., 2014*; *Piyush et al., 2019*; *Rai et al., 2019*; *Srivastava et al., 2021*; *Upadhyay et al., 2023*). Notably, when compared to clobetasol, curcumin demonstrated better efficacy in one of the studies (*Hazarey, Sakrikar & Ganvir, 2015*) but worse efficacy in another study (*Upadhyay et al., 2023*). In contrast, aloe vera gel (*Nerkar Rajbhoj et al., 2021*) and lycopene (*Saran et al., 2018*) were found to be superior to curcumin in terms of outcomes. Regarding side effects, eight studies reported the absence of side effects (*Yadav et al., 2014*; *Hazarey, Sakrikar & Ganvir, 2015*; *Ara et al., 2018*; *Piyush et al., 2019*; *Srivastava et al., 2021*; *Nerkar Rajbhoj et al., 2021*; *Adhikari et al., 2022*; *Upadhyay et al., 2023*), and one study reported flushing on the face and palms in one of the participants. The other patient in this study complained of abdominal discomfort, nausea, and stomach upset (*Rai et al., 2019*). These side-effects were not reported in the other two studies (*Saran et al., 2018*; *Namratha et al., 2023*).

### OLP

A significant improvement in OLP upon curcumin treatment was demonstrated in all studies, suggesting that curcumin was comparable to triamcinolone paste (*Kia et al., 2015*; *Thomas et al., 2017*), betamethasone local steroid lotion (*Nosratzehi et al., 2018*), and prednisolone (*Kia et al., 2020*). No side effects were observed in three studies (*Thomas et al., 2017*; *Nosratzehi et al., 2018*; *Kia et al., 2020*) and insignificant side effects were reported in the remaining one study (*Kia et al., 2015*).

### OLK

Clinical and histopathological evaluations indicated a significantly better treatment response to curcumin in OLK, but some participants were lost to follow-up owing to adverse effects (*Kuriakose et al., 2016*).

## Meta-analysis results

A meta-analysis was conducted to compare the efficacy of curcumin with that of the control, including placebo and conventional therapies, and a sub-analysis was performed to assess the improvement in various aspects of the three OPMDs. There were too few studies on OLK that met the inclusion criteria to be combined. Considering the substantial heterogeneity induced by various methods of drug administration, a subgroup analysis was conducted. Nonetheless, remarkable heterogeneity was noted in the included studies, which may have been owing to the study design, formulation of curcumin, various drugs in the active control groups, and participant health status. Owing to the high degree of

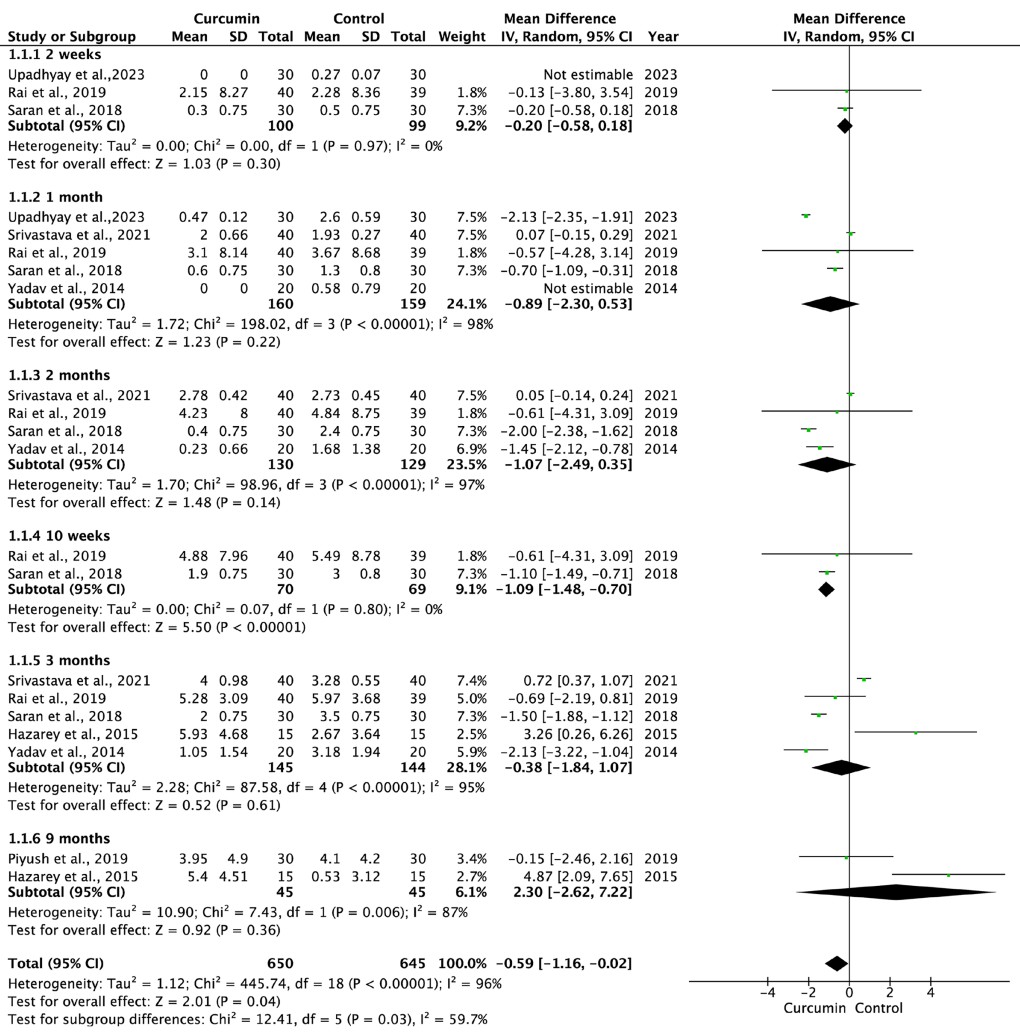

**Figure 4** Forest plots for treatment efficacy of curcumin and active control on mouth opening of oral submucous fibrosis (*Upadhyay et al., 2023*; *Srivastava et al., 2021*; *Rai et al., 2019*; *Piyush et al., 2019*; *Hazarey, Sakrikar & Ganvir, 2015*; *Saran et al., 2018*; *Yadav et al., 2014*).

heterogeneity within the groups, a random-effects model was employed to analyse the outcome indicators, including burning sensation and mouth opening. Unfortunately, trials for the systematic use of curcumin in OLP and topical application in OSF were insufficient to be combined, leading to an incomplete meta-analysis.

## OSF

Baseline OSF data were comparable between the curcumin and active control groups (Fig. S1). Additionally, the pooled data of seven trials showed a slightly lower efficacy of curcumin compared to that of the positive control groups in improving mouth opening: at 2 weeks ($I^2 = 0.00\%$, $P = 0.30$; SMD: −0.20, 95% CI [−0.58 to 0.18]), at 1 month ($I^2 = 98\%$, $P = 0.22$; SMD: −0.89, 95% CI [−2.30 to 0.53]), at 2 months ($I^2 = 97\%$, $P = 0.14$; SMD: −1.07, 95% CI [−2.49 to 0.35]), at 10 weeks ($I^2 = 0.00\%$, $P < 0.00001$; SMD: −1.09 95% CI [−1.48 to −0.70]), at 3 months ($I^2 = 95\%$, $P = 0.61$; SMD: −0.38, 95% CI [−1.84 to 1.07]),

A

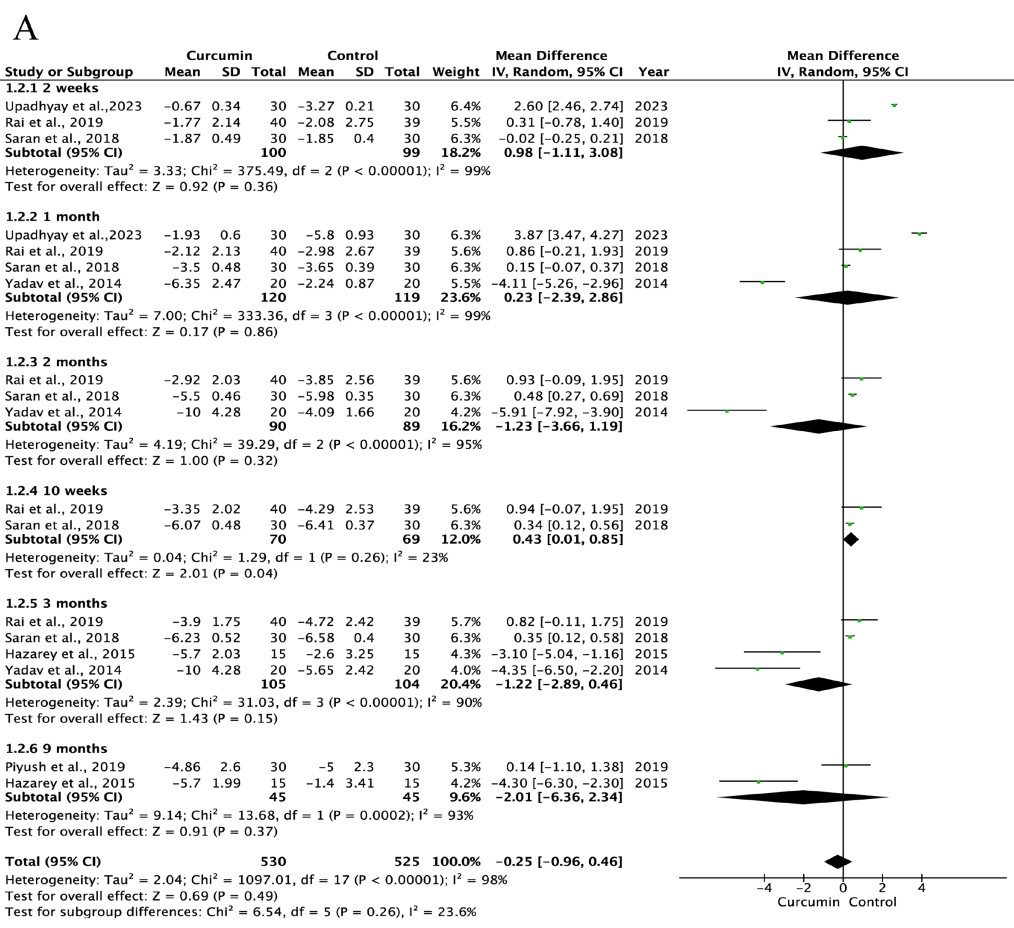

B

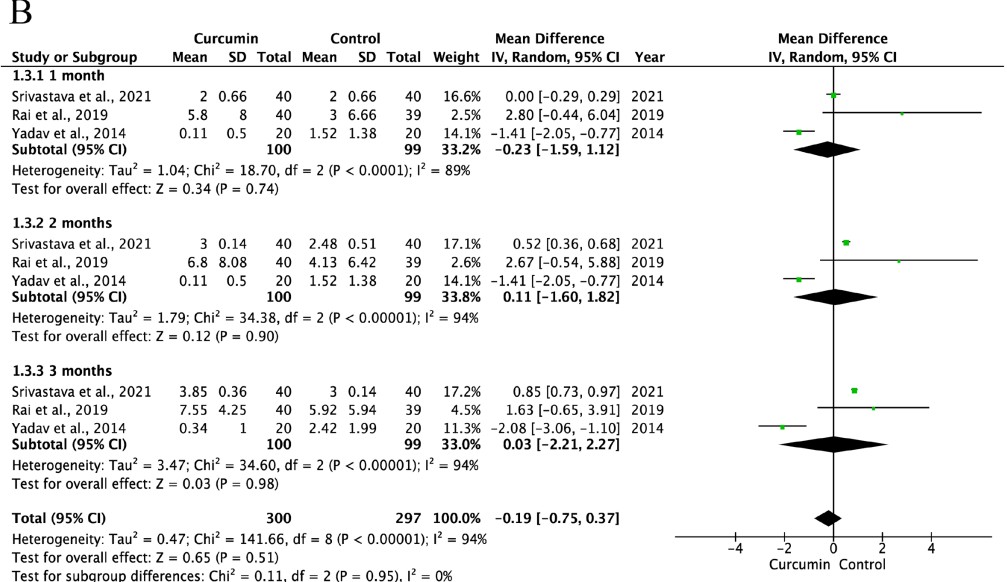

**Figure 5  Forest plots for therapeutic efficacy of curcumin and active control on pain amelioration (A) and tongue protrusion (B) of oral submucous fibrosis (*Upadhyay et al., 2023*; *Srivastava et al., 2021*; *Rai et al., 2019*; *Piyush et al., 2019*; *Hazarey, Sakrikar & Ganvir, 2015*; *Saran et al., 2018*; *Yadav et al., 2014*).**

A

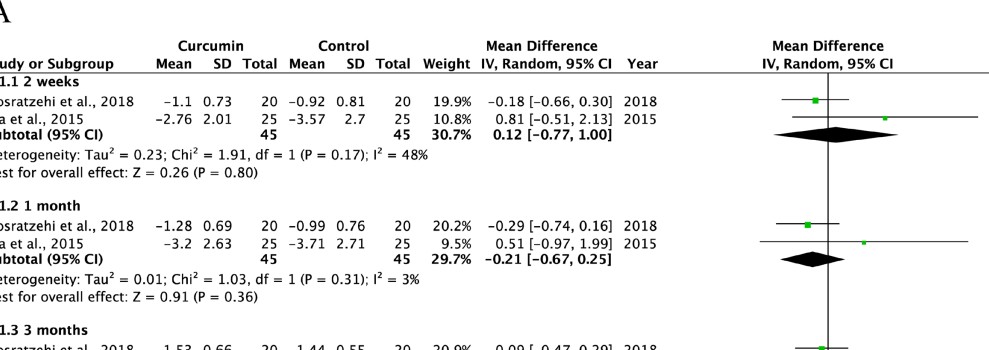

B

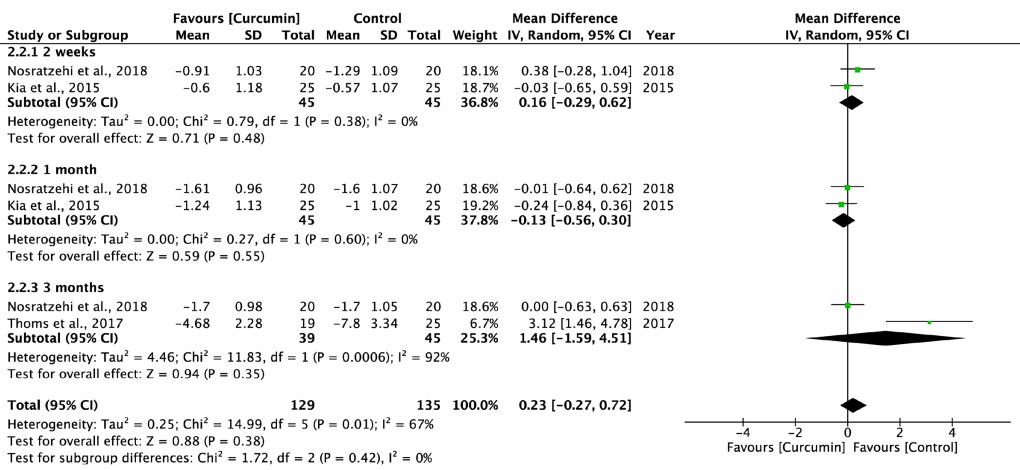

**Figure 6** Forest plots for managing efficacy of curcumin and active control on reducing pain (A) and clinical improvement (B) of oral lichen planus (*Nosratzehi et al., 2018*; *Kia et al., 2015*; *Thomas et al., 2017*).

and at 9 months ($I^2 = 87\%$, $P = 0.36$; SMD: 2.30, 95% CI [−2.62 to 7.22]) (Fig. 4). The pooled data from six studies showed curcumin had no statistically significant differences compared to control groups with respect to the pain reduction (lycopene and clobetasol in two studies, and antioxidant and dexamethasone in one study respectively): at 2 weeks ($I^2 = 99\%$, $P = 0.36$; SMD: 0.98, 95% CI [−1.11 to 3.08]), at 1 month ($I^2 = 99\%$, $P = 0.86$; SMD: 0.23, 95% CI [−2.39 to 2.86]), at 2 months ($I^2 = 95\%$, $P = 0.32$); SMD: −1.23, 95% CI [−3.66 to 1.19], at 3 months ($I^2 = 90\%$, $P = 0.15$; SMD: −1.22, 95% CI [−2.89 to 0.46]), and at 9 months ($I^2 = 93\%$, $P = 0.37$; SMD: −2.01, 95% CI [−6.36 to 2.34]) (Fig. 5A). Three trials showed that curcumin had no statistical difference compared to the control group in the improvement of tongue protrusion at 1 month ($I^2 = 89\%$, $P = 0.74$; SMD: −0.23, 95% CI

[−1.59 to 1.12]), 2 months ($I^2$ = 94%, $P$ = 0.90; SMD: 0.11, 95% CI [−1.60 to 1.82]), and 3 months ($I^2$ = 94%, $P$ = 0.98; SMD: 0.03, 95% CI [−2.21 to 2.27]) (Fig. 5B).

### OLP

The baseline OLP data were comparable between the curcumin and active control groups (Fig. S2). Three trials indicated that the efficacy of topical curcumin was equivalent to that of the positive control group, including steroids, in alleviating pain at 2 weeks ($I^2$ = 48%, $P$ = 0.80; SMD: 0.12, 95% CI [−0.77 to 1.00]), at 1 month ($I^2$ = 3%, $P$ = 0.36; SMD: −0.21, 95% CI [−0.67 to 0.25]), and at 3 months ($I^2$ = 95%, $P$ = 0.39; SMD: 0.70, 95% CI [−0.88 to 2.29]) (Fig. 6A). With respect to the clinical refinement of the lesions, there was a comparable effect between the curcumin and control groups: at 2 weeks ($I^2$ = 0.00%, $P$ = 0.48; SMD: 0.16, 95% CI [−0. 29 to 0.62]), at 1 month ($I^2$ = 0%, $P$ = 0.55); SMD: −0.13, 95% CI: [−0.56 to 0.30], and at 3 months ($I^2$ = 92%, $P$ = 0.35; SMD: 1.46, 95% CI [−1.59 to 4.51]) (Fig. 6B).

## Publication bias

Because only a limited number of studies were included for each type of OPMDs, publication bias could not be determined.

## DISCUSSION

Considering the significantly reduced quality of life due to OPMDs and the lack of an effective management approach with fewer adverse effects, there is an urgent need to identify an optimal alternative management strategy. Although several systematic reviews and meta-analyses have been conducted to analyse the role of curcumin in improving the symptoms of OPMDs, the results vary (Table 2) (*Ara et al., 2016*; *Al-Maweri, 2019*; *White, Chamberlin & Eisenberg, 2019*; *Rai et al., 2021*, *2023*; *Moayeri et al., 2024*; *Shao, Miao & Wang, 2024*). *Al-Maweri (2019)* performed a systematic review incorporating six clinical trials to evaluate the efficacy of curcumin in OSF, and the results showed that curcumin is a promising treatment option for the management of OSF. In addition, *Rai et al. (2021)* conducted a systematic evaluation and meta-analysis of turmeric in the management of OSF; 11 trials were included in this study, while only three trials were included in the meta-analysis. However, only the effect on improvement of mouth opening was validated, while evidence for other effects could not be achieved owing to the poor methodological quality of the studies. In addition, different interventions for OSF were compared through a network meta-analysis in a recent study (*Rai et al., 2023*). Although curcumin was indicated to be the most effective treatment for tongue protrusion, more attention was paid to the combinatorial therapy rather than the individual efficacy of curcumin. Regarding the efficacy of curcumin for OSF, a recent systematic review and meta-analysis demonstrated comparable effects of systemic or topical curcumin as control approaches (*Shao, Miao & Wang, 2024*); however, the efficacy of curcumin could not be evaluated individually, as it was administered together with other medicines. In addition, a systematic review evaluating the efficacy of curcumin for OLP was subsequently conducted by another group, who suggested that curcumin was unlikely to replace topical corticosteroids in most patients (*White, Chamberlin & Eisenberg, 2019*). Next, one recent study by *Moayeri et al.*

**Table 2 Summary of previously published systemic reviews or meta-analyses assessing the therapeutic efficacy of curcumin for OPMDs.**

| Authors | Year | Type of OPMDs | Study type | Interventions | Comparisons | Conclusion |
|---|---|---|---|---|---|---|
| *Ara et al. (2016)* | 2016 | OPMDs | Meta-analysis | Curcumin in systemic and topical way | NI | Insufficient evidence is presented to evaluate efficacy of curcumin for OPMDs specially on OSF. |
| *Al-Maweri (2019)* | 2019 | OSF | Systematic review | Curcumin in systemic way | Any medical interventions and/or placebo control | Curcumin is a promisingly effective treatment option for the management of OSF. |
| *Rai et al. (2021)* | 2021 | OSF | Systemic review and meta-analysis | Curcumin in systemic and topical way | Lycopene capsules, clobetasol propinonate, nigella sativa, multinal tablets | Curcumin is a potentially effective treatment choice for the management of OSF. |
| *Rai et al. (2023)* | 2023 | OSF | Network meta-analysis | Any medicinal interventions for OSF | Any other medicinal intervention or placebo for OSF | The combined treatment with steroid, hyaluronidase, and antioxidant was ranked as the most effective for improvement of mouth opening and burning sensation. And the single application of curcumin was found as the most effective means for improving tongue protrusion. |
| *Shao, Miao & Wang (2024)* | 2024 | OSF | Systemic review and meta-analysis | Curcumin in systemic and topical way; curcumin combining with other drugs (aloe vera, dexamethasone with hyaluronidase, steroids). | Steroids, lycopene, antioxidants, dexamethasone and hyaluronidase, hydrocortisone and hyaluronidase | Curcumin may have potential in improving mouth opening, alleviating oral burning sensation, and improving tongue protrusion of OSF. |
| *White, Chamberlin & Eisenberg (2019)* | 2019 | OLP | Systematic review | Curcumin in systemic and topical way | Placebo, corticosteroids | Topically applied curcumin in particular shows promising preliminary data but is still inferior to topical corticosteroids as the modality of choice for most patients. |
| *Moayeri et al. (2024)* | 2024 | OLP | Systemic review and meta-analysis | Curcumin in systemic and topical way | Placebo, steroids | Curcumin had no significant effect on erythema, lesion size, and pain of OLP compared to the control groups. However, curcumin was more effective in reducing pain in non-randomized trials and in trials with a treatment duration of 2 weeks. |

**Note:**

Abbreviations: OSF, oral submucous fibrosis; OLP, oral lichen planus; OPMDs, oral potentially malignant disorders; NI, non-informed.

*(2024)* assessed the effects of curcumin on relieving OLP symptoms by employing meta-analysis in 10 included studies, demonstrating that curcumin (topical application) treatment for a duration of 2 weeks shows better performance in reducing pain compared with control sets. Notably, in this study, topical and systemic application of curcumin, as well as positive and placebo controls were analysed together, resulting in relatively high heterogeneity (*Moayeri et al., 2024*). Although a meta-analysis in 2016 sought to assess the efficacy of curcumin in OPMDs, only five trials were included, which resulted in incomplete and unconvincing results (Table 2) (*Ara et al., 2016*). Conversely, our study

explored the efficacy of curcumin alone in the treatment of OPMDs. Compared with the previously published meta-analyses mentioned above (*Ara et al., 2016*; *Rai et al., 2021*; *Moayeri et al., 2024*), our study indicated that curcumin may not only relieve pain and improve tongue protrusion but also refine the mouth opening in OSF, although slightly inferior to the positive control groups, which was not revealed in the previous analysis. In addition, we concluded that curcumin could potentially achieve significant improvements in pain and erosion associated with OLP. The differences between our study and previous studies are as follows: firstly, the efficacy of curcumin was evaluated for OPMDs and not for OSF or OLP alone; secondly, we included studies until March 2024; thirdly, a sub-analysis of different outcome indicators for each disease was performed to ensure more comprehensive results; fourthly, we not only performed subgroup analysis regarding its topical or systemic application but also distinguished positive from placebo controls; and finally, varied inclusion or exclusion criteria were adopted in our study.

Sixteen RCT (OSF, 11; OLP, four; and OLK, one) were included in this review, and the results indicated that curcumin may be effective in treating OPMDs, including OSF, OLP, and OLK. Owing to the relatively strict inclusion criteria and subgroup analysis in our study, the number of studies included in each analysis was slightly insufficient, preventing an assessment of publication bias. Nonetheless, the risk of bias results were relatively convincing, indicating nearly comparable effects of curcumin and traditional therapies in the management of OLP and OSF. However, in studies adopting a placebo as a control group, topical application of curcumin for OSF (*Ara et al., 2018*; *Nerkar Rajbhoj et al., 2021*; *Adhikari et al., 2022*) and systemic use of curcumin for OLP (*Kia et al., 2020*), the number of trials was too small to be combined for a meta-analysis. Only one study of OLK met the inclusion criteria, and the others were excluded for the lack of a control group (*Cheng et al., 2001*; *Deb et al., 2022*; *Kapoor & Arora, 2019*). However, all these studies demonstrated the potential of curcumin for treating OLK to reduce lesion size and chemoprevention of its cancerisation.

The key finding of our study was the equivalent efficacy of curcumin as a corticosteroid in alleviating pain in OSF and OLP patients (Table 3), suggesting that curcumin is beneficial in improving patients' quality of life while reducing the risks triggered by corticosteroids. Similarly, several studies have demonstrated similar efficacy of curcumin in alleviating pain (*Vaughn, Branum & Sivamani, 2016*; *Salehi et al., 2019*; *Girisa et al., 2021*; *Patil et al., 2022*), and one previous meta-analysis revealed no marked difference between the efficacy of curcumin and prednisolone for managing OLP (*Zeng et al., 2022*). Curcumin ameliorates pain through its anti-inflammatory and antioxidant effects. Its anti-inflammatory function appears to be regulated by the inhibition of tumour necrosis factor-α, interleukins-6, and interleukins-8 (*Fernández-Lázaro et al., 2020*). In addition, curcumin may modulate sirtuins that activate the expression of inflammatory markers (*Ungurianu, Zanfirescu & Marginǎ, 2022*). Interestingly, one of the included studies found significantly lower inflammatory biomarkers in OLP patients treated with curcumin than in those receiving a placebo, confirming the role of curcumin in regulating inflammation

**Table 3 The efficacy of curcumin in the management of various aspects of individual OPMDs.**

| Type of OPMDs | Drug administration | Aspects | Efficacy |
|---|---|---|---|
| OSF | Systemic | Mouth opening | Curcumin showed lower efficacy compared to positive control groups in improving mouth opening of OSF. |
| | Systemic | Pain amelioration | Curcumin could refine the pain of OSF, showing potentially comparable effects as the positive controls including steroids. |
| | Systemic | Tongue protrusion | Curcumin could refine the tongue protrusion of OSF, showing potentially comparable effects as the positive controls including steroids. |
| OLP | Topical | Pain amelioration | Curcumin might be able to palliate the pain of OLP patients, showing potentially comparable effects as the positive controls including steroids. |
| | Topical | Clinical improvement | Curcumin might be able to promote clinical healing of OLP patients, showing potentially comparable effects as the positive controls including steroids. |
| OLK | Systemic | Clinical and histologic response | Clinical and histopathologic evaluations indicated significantly better treatment response of curcumin than placebo for OLK. |

**Note:**
Abbreviations: OSF, oral submucous fibrosis; OLP, oral lichen planus; OPMDs, oral potentially malignant disorders; OLK, oral leukoplakia.

(*Chainani-Wu et al., 2012*). Additionally, curcumin exhibits antioxidant effects by activating nuclear respiratory factors (*Dai et al., 2018*).

Our study also supports the relatively active role of curcumin in improving the clinical symptoms of OPMDs, such as the alleviation of erythema or erosion in OLP and improvement of tongue protrusion in OSF (Table 3), consistent with previous systematic reviews (*Girisa et al., 2021*; *Zeng et al., 2022*). Specifically, curcumin may be able to prevent OSF development and improve mouth opening by regulating the expression of OSF pathophysiological factors including transforming growth factor-β and p53 (*Gupta et al., 2017*). Notably, despite the slightly lower efficacy of curcumin in improving mouth opening in the OSF group than in the positive control group in our study, curcumin may serve as an optimal alternative medicine when steroids are not administered. This conclusion was partially supported by a previous network meta-analysis in which the combination of steroids, hyaluronidase, and antioxidants was reported as the most effective means to improve mouth opening (*Rai et al., 2023*).

Preventing malignant transformation is another important objective in managing OPMDs. Inhibiting the onset and progression of epithelial dysplasia is regarded as the optimal way to prevent malignant transformation in OPMDs (*Warnakulasuriya, 2020*). However, the current data are insufficient to conduct a relevant analysis. One study included in our systematic review revealed that OLK showed a significantly better response to curcumin treatment (Table 3) (*Kuriakose et al., 2016*). The underlying mechanism is mainly attributed to its anticancer potential and protection of cells from oxidative damage (*Menon & Sudheer, 2007*; *Giordano & Tommonaro, 2019*; *Zia et al., 2021*). Additional well-designed trials are required to validate the ability of curcumin in inhibiting the malignant transformation of OPMDs.

Although corticosteroids are commonly used to treat OSF and OLP, their long-term use is associated with certain adverse effects. It is worth noting that our study indicated that

curcumin demonstrates promising potential to be an alternative to corticosteroids in some contexts.

Our study had certain limitations. Firstly, numerous trials were excluded owing to the combined use of curcumin with other medications to treat OPMDs, which resulted in a deficient number of studies in each analysis. Secondly, significant heterogeneity was found in the meta-analysis in terms of study design, formulation of curcumin, drug in the control groups and its formulation, health education, follow-up period, and age and sex of the participants. To decrease substantial heterogeneity, a subgroup analysis of different approaches of drug administration was conducted, which further reduced the sample size. Thirdly, the risk of bias showed methodological shortcomings in some of the included trials, particularly in the selection of reported results. Finally, only one study on OLK was included, resulting in no definitive conclusion regarding the efficacy of curcumin in OLK.

## CONCLUSION

Overall, our study findings indicated that curcumin could potentially improve the control of pain and erosion in OLP and may also refine pain and tongue protrusion in OSF, showing potentially comparable effects to the positive controls, including steroids. Curcumin demonstrated relatively lower efficacy than the positive control in improving mouth opening in OSF patients. In addition, considering the comparatively marked heterogeneity of the included trials, more high-quality RCTs with larger sample sizes, sufficient follow-up periods, standardised curcumin formulations, and more comprehensive types of OPMDs are required to assess the efficacy of curcumin in managing other OPMDs and controlling their malignant transformation, thus providing more robust evidence for its clinical application.

## ACKNOWLEDGEMENTS

We would like to thank the authors of these original studies on this topic included in our analysis.

### Funding

This work is supported by the National Natural Science Foundation of China (82272899, 81902782, 82203180), the Research Funding from West China School/Hospital of Stomatology Sichuan University (No. RCDWJS2022-16), the 14th special grant from China Postdoctoral Science Foundation (2021T140484), the Postdoctoral Research Funding of Sichuan University (2022SCU12132), the Research and Exploration Program of West China Hospital of Stomatology of Sichuan University (No. RD-02-202204), the Key Research Program of Sichuan Provincial Science and Technology Agency (2023YFS0127) and the CAMS Innovation Fund for Medical Sciences (CIFMS,2019-I2M-5-004). The funders had no role in study design, data collection and analysis, decision to publish, or preparation of the manuscript.

## Grant Disclosures

The following grant information was disclosed by the authors:
National Natural Science Foundation of China: 82272899, 81902782, 82203180.
West China School/Hospital of Stomatology Sichuan University: RCDWJS2022-16.
14th special grant from China Postdoctoral Science Foundation: 2021T140484.
Sichuan University: 2022SCU12132.
Research and Exploration Program of West China Hospital of Stomatology of Sichuan University: RD-02-202204.
Key Research Program of Sichuan Provincial Science and Technology Agency: 2023YFS0127.
CAMS Innovation Fund for Medical Sciences: CIFMS,2019-I2M-5-004.

## Competing Interests

The authors declare that they have no competing interests.

## Author Contributions

- Wenjin Shi performed the experiments, analyzed the data, prepared figures and/or tables, authored or reviewed drafts of the article, and approved the final draft.
- Qiuhao Wang performed the experiments, analyzed the data, prepared figures and/or tables, and approved the final draft.
- Sixin Jiang performed the experiments, prepared figures and/or tables, and approved the final draft.
- Yuqi Wu analyzed the data, authored or reviewed drafts of the article, and approved the final draft.
- Chunyu Li performed the experiments, authored or reviewed drafts of the article, and approved the final draft.
- Yulang Xie analyzed the data, authored or reviewed drafts of the article, and approved the final draft.
- Qianming Chen conceived and designed the experiments, authored or reviewed drafts of the article, and approved the final draft.
- Xiaobo Luo conceived and designed the experiments, authored or reviewed drafts of the article, and approved the final draft.

## Data Availability

    This is a systematic review/meta-analysis.

## Supplemental Information

Supplemental information for this article can be found online at http://dx.doi.org/10.7717/peerj.18492#supplemental-information.

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
