# Peer review of "Evaluating the efficacy of curcumin in the management of oral potentially malignant disorders: a systematic review and meta-analysis"

_PeerJ, doi:10.7717/peerj.18492_

## Round 0.1 · original submission · Major Revisions

Please revised as per the comments from the reviewers.

·

Basic reporting

1. Results of the meta analysis do not match with the authors conclusion on the efficacy of curcumin in the management on OSF or OLP.

Lines 294-295: 294 the pooled data of 7 trials showed a slightly lower efficacy of curcumin compared to positive control groups in improving mouth opening:
Line 311: The data at baseline of OLP was comparable between curcumin groups and active control groups

Despite above findings you conclude: Our study also supported the active role of curcumin in improving the clinical symptoms of OPMDs, such as the alleviation of erythema or erosion of OLP and the improvement of mouth opening restriction of OSF,

2, In Table 2, in the last column (main results) no data are given to come to the stated conclusion.

3, It is not clear what you say in lines 337-338 "uncompleted published bias was induced"

4. The results of previous systematic reviews stated in the introduction (lines 120-137) should be moved to the discussion

5. The objective of the study is not clear

6. Lines; 69 and 101. The term "so forth" should not be used in writing .

7. Line 72: the term "betel nut" should not be used in scientific writing . Instead Use "areca nut"

8. Hazarey et al RCT (your ref 37) on OSF is missing in Table 2

9. English grammar need improving. Please get help from an English writer,

Experimental design

It is not clear whether the authors used a random-effect model or a fixed-effect model for this systematic review-meta analysis.

Validity of the findings

Results do not match with the authors' conclusion.

Additional comments

English needs improvement to suit an international readership.

·

Basic reporting

The article is clear, unambiguous, with English professional used

Experimental design

Origial research and useful for all categories of medical and destist; research question is well defined, relevant, and try to make more clear the efficacy of Curcumin. Very rigorous investigation and meta analysys wery well done. optimum method and well described

Validity of the findings

This research has a very good impact and the data used are robust, statistically optimum sound and controlled
Conclusions are well stated, linked to the original research question and limited to supporting results

Additional comments

very well done manuscript and very simply to understand also for not special addicts

·

Basic reporting

no comment

Experimental design

no comment

Validity of the findings

1) Kindly incorporate the pre and post changes in the clinical figure shown with all the respective lesions.
2) Mention the effectiveness of curcumin alone and curcumin with steroids, both topically as well as systemically. Compare and contrast.
3) Stratify the OPMDS and present the table for the same with context to the present topic.
4) Mention different forms of curcumin and its medical application in short.

Additional comments

no comments

---

## Round 0.2 · accepted · Accept

According to the reviewer reports, and my own evaluation, this paper can be accept for publication

·

Basic reporting

no comment

Experimental design

no further comment

Validity of the findings

no further comment

Additional comments

none

·

Basic reporting

no
comments

Experimental design

no
comments

Validity of the findings

no
comments

Additional comments

no
comments